Investigating leaf beetles (Coleoptera, Chrysomelidae) on the west coast islands of Sabah via checklist-taking and DNA barcoding

http://orcid.org/0000-0003-1981-3928 Yeong Kam-Cheng 1 kamchengyeong@gmail.com
Takizawa Haruo 2
Liew Thor-Seng 1 3
1 Institute for Tropical Biology and Conservation, Universiti Malaysia Sabah , Kota Kinabalu, Sabah , Malaysia
2 Nodai Research Institute, Tokyo University of Agriculture , Tokyo , Japan
3 Small Islands Research Centre, Universiti Malaysia Sabah , Kota Kinabalu, Sabah , Malaysia
Gillespie Joseph
Electronic publication date: 2018 Oct 25
Publication date: 2018
Volume: 6
Electronic Location ID: e5811
Received 2018 Mar 29; Accepted 2018 Sep 23
Copyright: © 2018 Yeong et al.
Copyright year: 2018
Copyright holder: Yeong et al.
License: This is an open access article distributed under the terms of the Creative Commons Attribution License, which permits unrestricted use, distribution, reproduction and adaptation in any medium and for any purpose provided that it is properly attributed. For attribution, the original author(s), title, publication source (PeerJ) and either DOI or URL of the article must be cited.
License URL: https://creativecommons.org/licenses/by/4.0/

Keywords: Barcoding of Life Data system, Island biodiversity, Colour polymorphism, Species distribution, Cryptic species, Chrysomelid beetles, Polymorphic species

Funding: The Universiti Malaysia Sabah under the UMSGreat research GUG0014-SG-M-1/2016 This work was supported by the Universiti Malaysia Sabah under the UMSGreat research grant (GUG0014-SG-M-1/2016). The funders had no role in study design, data collection and analysis, decision to publish, or preparation of the manuscript.

==============================
Sabah is a province of Malaysia located on the northern part of the island of Borneo. Most of the leaf beetle fauna studies from this region conducted over the past 15 years have focussed on the mainland habitats while the leaf beetle fauna from island habitats (ca. 500 islands) have largely been overlooked. This study looks into the leaf beetle fauna of 13 small satellite islands off the west coast of Sabah. All specimens were first sorted into morpho-species operational taxonomic unit (OTU) before being identified to species rank where possible based on morphological characters and species names assigned when the specimens fitted the description of species in the literature. We collected 75 OTUs from 35 genera and five subfamilies according to morphology, 12 of which were identifiable to species level. In addition, the DNA barcode for each OTU was cross checked with records in GenBank and Barcoding of Life Data system (BOLD) to verify their identity. The number of species recorded was reduced from 12 species and 63 OTUs (total 75 OTUs) to 12 species and 56 OTUs (total 68 OTUs) after removal of the colour polymorphic species based on DNA barcode analyses. Pulau Gaya has the highest species richness and Pulau Sulug has the lowest species richness. A total of 64 Barcode Index Numbers consisting of 101 DNA barcodes were obtained from the 12 leaf beetle species and 48 OTUs. Based on the DNA barcode analyses, it was possible to confirm several polymorphic OTUs and cryptic species. The mean intraspecific and interspecific genetic divergence were determined as 0.77% and 16.11%, respectively. DNA barcodes of this study show a low similarity with records in GenBank and BOLD, highlighting the lack of representation and the urgency of studying leaf beetles from this region. The study provides the first documentation of leaf beetle fauna from island habitats of Sabah and the first DNA barcoding data for leaf beetles from this part of the world, with the next steps being larger scale sampling over a wider geographical scale for a better understanding of tropical arthropod diversity.

Introduction

Chrysomelidae Latreille, 1802 is one of the most diverse beetle families, with 35,000–60,000 species around the world (Splipnski, Leschen & Lawrence, 2011; Jolivet, 2015). The study of leaf beetle fauna in Borneo started in the 19th century, with the first valid species described by Suffrian (1854). A brief history of leaf beetle studies in Borneo is discussed in The Leaf Beetle of Borneo by Mohamedsaid, Salleh & Hassan (1992). Although Borneo is recognized as one of the world’s most biodiverse islands, taxonomic research on Borneo leaf beetles has been limited to a few publications. As of 2004, 635 species of leaf beetle had been recorded in Borneo (Mohamedsaid, 2004). Researchers are currently in a race against time to study the Bornean leaf beetles due to the high deforestation rates in the region (Sodhi et al., 2010).

Over the past a decade and a half, the number of leaf beetle species in Borneo has increased significantly, with more than 100 new species originating from Sabah (Takizawa, 2005, 2011, 2012, 2013, 2014, 2017; Mohamedsaid, 2006a, 2010; Beenen, 2007; Doberl, 2007; Medvedev, 2007, 2009, 2010, 2013, 2016a, 2016b; Borowiec, 2009; Moseyko, 2012; Borowiec, Takizawa & Swietojańska, 2013; Mahadimenakbar & Takizawa, 2013; Bezděk, Romantsov & Medvedev, 2014; Medvedev & Romantsov, 2014, 2015, 2017a; Takizawa & Mohamedsaid, 2015). Most of these new species were discovered in mainland habitats on the west coast of Sabah. Although Sabah has the highest number of islands in Malaysia, leaf beetle species diversity on these islands has been little explored.

The above mentioned recent taxonomic works are based on morphological characteristics (Mohamedsaid, 2006a; Moseyko, 2012; Medvedev & Romantsov, 2017b) and few studies have utilized the molecular approaches to infer the phylogeny of leaf beetles (Kishimoto-Yamada et al., 2013; Crampton-Platt et al., 2015). Using this conventional taxonomic approach alone is challenging because sexual dimorphism and colour pattern variants or phenotypic polymorphism are common, especially variables within the subfamily Galerucinae (Crowson, 1981; Chaboo, 2007; Prado, 2013; Gómez-Zurita et al., 2016). Consequently, DNA barcoding has been added to the taxonomist’s toolkit in order to complement the species identifications that are based on morphological characters (Hebert et al., 2003; Pentinsaari, Hebert & Mutanen, 2014; Gómez-Zurita et al., 2016). To date, there are 73 records of leaf beetles with 15 Barcode Index Numbers (BINs) from Malaysia registered in the Barcoding of Life Data system (BOLD), but none of these records are from Sabah or Borneo.

For all the reasons stated above, it is timely to (1) document the species richness of leaf beetles from different habitats to obtain the general pattern of Bornean leaf beetle diversity, and (2) generate DNA barcodes for known and newly described Bornean leaf beetle species to serves as supplementary information for the morphological identification and building up of the DNA barcodes reference database and to enable rapid species identification. Therefore, this study serves as part of the effort by the second author, Haruo Takizawa towards the above mentioned goals through documentation of the species richness of leaf beetles in Sabah.

Materials and Methods

Leaf beetle sampling and processing

Standard plot sampling was carried out between January 2016 and March 2017 on the 13 islands along the west coast of Sabah (Fig. 1; Table 1) under research permit from Sabah Park (TTS/IP/100-6/2 Jld.4 (49)) and permission from Sapangar Naval Base (MWL2.100-2/2/3-(9)). The number of sampling plot varies among the sampled islands according to their respective island area (Table 1). In each 20 × 20 m plot, a manual hand-picking search for leaf beetles and 200 sweeps of shrubs and herbaceous vegetation were conducted using an entomological sweep net (Sánchez-Reyes, Niño-Maldonado & Jones, 2014) over two person-hours. Leaf beetles from each plot were kept in separate Falcon Tubes and brought to the laboratory for further processing. It should be noted that specimens from outside the plots were also collected.

Figure 1 The 13 Sabah west coast islands selected and sampled in this study.

(A) Sabah overview map with the location of each island group (highlighted in red and selected in box); (B) Sapangar bay island group; (C) Tunku Abdul Rahman Park; (D) Pulau Dinawan and Pulau Mantukod; (E) Pulau Tiga; (F) Pulau Mengalum. Pulau = Island.

Table 1 Coordinate, area (km2), distance from nearest mainland (km) and number of plot(s) on the 13 sampled west coast islands of Sabah.

Island name (Pulau)	Latitude	Longitude	Area (km2)	Distance from nearest mainland (km)	Number of plot(s)	
1. Pulau Dinawan	5.8472	115.9907	0.2603	3.0116	5	
2. Pulau Gaya	6.0176	116.0316	14.3166	1.4657	16	
3. Pulau Mamutik	5.9666	116.0137	0.0563	3.2546	3	
4. Pulau Mantukod	5.8379	116.0129	0.0968	1.0626	3	
5. Pulau Manukan	5.9753	116.0012	0.4402	4.3244	9	
6. Pulau Mengalum	6.2001	115.5968	5.1640	53.8144	7	
7. Pulau Peduk	6.0873	116.0963	0.0052	0.4369	1	
8. Pulau Sapangar	6.0674	116.0738	1.3188	2.3408	8	
9. Pulau Sapi	6.0095	116.0061	0.1877	6.9039	3	
10. Pulau Sulug	5.9599	115.9933	0.1261	5.1081	3	
11. Pulau Tiga	5.7235	115.6521	6.9860	10.1733	12	
12. Pulau Udar Besar	6.0794	116.0881	0.0369	1.4654	3	
13. Pulau Udar Kecil	6.0849	116.0942	0.0748	0.5930	2	

Leaf beetle specimens collected were first killed using 70% ethanol before identified under the microscope. All the specimens were first sorted and grouped into morpho-species operational taxonomic unit (OTU) before identified to species rank when possible by the second author based on morphological characteristics and literature available. For OTUs that were unable to be identified to species rank, these specimens were identified to genus rank when possible. The usage of the term ‘species’ in the present study referred to those determined species based on morphological characters and literature available while OTUs referred to both morpho-species and putative species determined through DNA barcoding. A few representative specimens of each species and OTU were selected and kept in absolute non-denatured ethanol under −20 °C for further DNA analysis. Photographs for dorsal and ventral habitus were taken for each species and OTU using a Leica Stereoscope M165C acquired with Leica DFC495 camera and Leica Application Suite ver.4.4.0. All the specimens were deposited in the BORNEENSIS collection at the Institute for Tropical Biology and Conservation, Universiti Malaysia Sabah. Specimen information was catalogued in the BORNEENSIS collection database under the accession number BOR/COL ####.

DNA extraction, PCR amplification and sequencing

DNA was extracted from one to three whole legs of the leaf beetles using Qiagen DNeasy Blood and Tissue Kits, following the manufacturer’s protocols. After that, all the DNA extracts were stored under –20 °C prior to PCR amplification.

The mitochondrial gene region, cytochrome c oxidase subunit I was PCR-amplified using universal primer LCO 1490 and HCO 2198 (Folmer et al., 1994). The 25 μl PCR reaction mixtures contained 2.5 μl of 10× GoTaq® PCR buffer with 15 mM MgCl2, 1.5 μl of 25 mM MgCl2, 0.5 μl of 2.5 mM dNTP mix, 0.5 μl of 10 pmol each primer, 0.25 μl of 5 u/μl Taq DNA polymerase, one μl DNA template and 18.25 μl ddH2O. PCR amplification was performed in Bio-Rad T100 Thermal Cycler following thermal cycling, an initial denaturation at 94 °C for 3 min, followed by 35 cycles of denaturation at 94 °C for 30 s, annealing at 47 °C for 45 s, extension at 72 °C for 60 s, and a final extension at 72 °C for 5 min. PCR products were checked for successful amplicon using the 1% agarose gel with TBE buffer. Successful PCR amplicons were sent to Genomics BioScience and Technology Co., Ltd. (Taipei, Taiwan) for sequencing.

Data analysis

Sequences were checked visually with Bioedit v7.1.9 (Hall, 1999). All the complete sequences were uploaded, registered and managed in the BOLD (Ratnasingham & Hebert, 2007) together with the information about taxonomy and collection, voucher deposition details, original sequence trace files and photographs of the specimens. Each sequence was assigned the BIN in BOLD (Ratnasingham & Hebert, 2013). Barcode Gap Analysis and distance summary for intraspecific and interspecific distance based on Kimura 2-parameter (K2P) distance model (Kimura, 1980) were performed in BOLD.

In addition, all sequences were crossed checked with the records in the National Center for Biotechnology Information GenBank and BOLD using the Basic local alignment search Tool (BLAST) (Altschul et al., 1990) in Geneious free trial v11.0.3 (Kearse et al., 2012) and identification tool in BOLD platform to search for similar DNA sequences in the database and to verify their tentative taxa identities. Resulting BLAST and BOLD top-hits for all the sequences are shown in Table S1.

DNA barcodes

The BINs for each specimen were listed. The intraspecific and interspecific distances of the species were generated using the sequence analysis in BOLD. For intraspecific distance, only species with more than one individual sequence were shown and for interspecific distance, only two or more species under the same genus were shown in the checklist. The ‘Mean’ represents the mean distance, ‘Max’ represents the maximum distance, and abbreviation ‘N/A’ represents data that are not available.

Species checklist

This checklist is comprised of information about the generic diagnosis of the genus: examined materials in BORNEENSIS, species distribution in west coast islands of Sabah, DNA barcode and general remarks on the species. The taxonomy nomenclature of the present study follows the Family-group names in Coleoptera (Bouchard et al., 2011) at the levels of family and subfamily, while at the levels of genus and species was adopted the Leaf Beetle genera (Seeno & Wilcox, 1982) and the Catalogue of the Malaysian Chrysomelidae (Mohamedsaid, 2004). Morpho-species that could not be identified to genus level were named after the subfamily (e.g. Galerucinae sp.). Photos of dorsal and ventral habitus for each of the species were included.

The generic diagnosis were done using literatures of the original description of the genus, other studies mentioning generic diagnosis characters, or the identification key to the genus level (Baly, 1860, 1865; Clark, 1865; Fairmaire, 1881; Jacoby, 1884, 1908; Lefevre, 1884; Sharp, 1904; Maulik, 1919, 1926, 1936; Laboissiere, 1933; Gressitt & Kimoto, 1963; Samuelson, 1969; Gressitt, 1969; Kimoto & Gressitt, 1979, 1981, 1982; Kimoto, 1989; Mohamedsaid, 1994; Medvedev, 2009; Hazmi & Wagner, 2010; Blanco & Konstantinov, 2013; Borowiec, Takizawa & Swietojańska, 2013; Reid & Beatson, 2015). Then, the distribution of each species on the sampled islands was summarized. Additional information on each of the species such as morphological characteristics, comparison with other species, sampled habitat, phenotypic variation and possible host plants were provided as remarks.

Results

Species diversity among leaf beetles

A total of 1,104 leaf beetle specimens were collected in this study, including 68 OTUs in 35 genera and five subfamilies after the removal of polymorphic species based on the DNA barcode analyses, in which 12 OTUs were identified to species level based on morphological characters. Of all the leaf beetle subfamilies collected, Galerucinae had the highest number of genera and species recorded (18 genera, 42 OTUs), followed by the subfamily Eumolpinae (nine genera, 16 OTUs), subfamily Cassidinae (five genera, seven OTUs), subfamilies Chrysomelinae (two genera, two OTUs) and Criocerinae (one genus, one OTUs). In terms of genera, genus Monolepta was the most speciose with 18 OTUs collected, followed by genus Hoplosaenidea with seven OTUs and genus Basilepta with six OTUs. In addition, Pulau Gaya has the highest species richness recorded amongst the sampled islands (42 OTUs), followed by Pulau Tiga (22 OTUs), Pulau Sapangar (18 OTUs), Pulau Dinawan and Pulau Sapi (nine OTUs), Pulau Mantukod and Pulau Manukan (eight OTUs), Pulau Mengalum (seven OTUs), Pulau Mamutik and Pulau Udar Besar (six OTUs), Pulau Udar Kecil (five OTUs), Pulau Peduk (four OTUs) and Pulau Sulug (two OTUs). The number of leaf beetle subfamilies, genera and OTUs collected at each island is summarized in Table 2.

Table 2 Number of subfamilies, genera, OTUs, BINs and shared BINs recorded on each west coast island of Sabah.

Pulau	Number of subfamilies	Number of genera	Number of OTUs	Number of BINs	Number of BINs shared with other islands	
Pulau Dinawan	3	6	9 [2]	8	6	
Pulau Gaya	4	23	42 [6]	36	23	
Pulau Mamutik	2	5	6 [2]	6	4	
Pulau Mantukod	2	5	8 [0]	7	6	
Pulau Manukan	3	6	8 [2]	7	6	
Pulau Mengalum	3	6	7 [2]	6	6	
Pulau Peduk	2	3	4 [1]	4	3	
Pulau Sapangar	3	11	18 [4]	15	13	
Pulau Sapi	3	6	9 [1]	8	7	
Pulau Sulug	2	2	2 [1]	2	2	
Pulau Tiga	4	15	22 [3]	20	13	
Pulau Udar Besar	1	3	6 [3]	6	5	
Pulau Udar Kecil	2	3	5 [0]	5	5	
Note:

Number in square brackets [] represent the number of OTUs identified to species rank based on morphological characters. Five Barcode Index Numbers for Nodina sp. were excluded due to the presence of cryptic diversity and morphologically hard to distinguish from one another.

DNA barcoding

Total genomic DNA of 75 morphologically identified leaf beetle species was extracted, but only 67 of these were successfully sequenced resulting in 100 barcode compliant sequences and one non-barcode compliant sequence. These 101 sequences were uploaded and assigned to 64 BINs in BOLD (available at: dx.doi.org/10.5883/DS-BCHRY18). Details of the sequenced leaf beetle species, number of specimens, and respective BINs are listed in the Supplementary file, Table S2. The differences between the number of successfully sequenced OTUs and the number of BINs suggests the presence of polymorphic species, which is confirmed by the sequence analyses in BOLD and reduced the species richness recorded in the present study (from 75 to 68 OTUs). Nonetheless, the congruence of species delimitation obtained through both morphological identification and DNA barcoding methods was relatively high (55 OTUs or 82.1%). The number of BINs for each island and number of BINs shared with other islands is summarized in Table 2. Smaller size islands tends to have a lower number of BINs and the majority of these BINs are shared with the other islands, while the larger size islands harbour more BINs and some of these were found to be exclusive to a particular island. A neighbour-joining tree was constructed based on these 101 sequences via BOLD (Fig. 2), to shows the relationship between these sequences.

Figure 2 Neighbour-joining tree for all the 101 analysed COI sequences (performed on the BOLD).

Clade highlighted in red colour represents leaf beetle species with phenotypic variation and clade highlighted in blue colour represents leaf beetle species with cryptic diversity.

From the sequence nucleotide composition analysis in BOLD, the average percentage of all the sequences G, C, A and T were 16.38% (14.58–18.13%), 17.19% (13.23–24.02%), 29.86% (27.11–33.13%) and 36.57% (30.66–40.99%), respectively. The overall mean AT content of the 101 sequences was 66.43% (57.85–71.66%) and strongly AT biased at the third codon position with mean AT content of 85.09% (63.64–96.51%). Intraspecific and interspecific K2P distances were easily distinguishable from each other, with overall means 0.77% (range 0–1.99%) and 16.11% (range 4.71–24.6%), respectively. Further details of the intraspecific and interspecific distances are available in the Tables S3 and S4.

All the 101 sequences submitted to GenBank through BLAST and identification tools in BOLD platform to search for identical results and the top-hit results are shown in the Table S1. The pairwise identity percentage of the 101 sequences with records in GenBank and BOLD range from 82.6% to 100% and from 84.01% to 100%, respectively. The majority of the GenBank and BOLD top-hit search record taxonomy are under the same order or family as the queried sequences, with some exceptional records (seven GenBank and BOLD records) classified under other insect orders (e.g. Order Lepidoptera) and other Coleoptera families (e.g. Zopheridae and Staphylinidae). These BLAST and BOLD top-hits results were summarized and grouped into two categories (≥90% and <90%) based on their pairwise identity percentages, as shown in Table 3.

Table 3 Summary of BLAST and BOLD top-hit results based on pairwise identity percentage (%).

Pairwise identity (%)	Taxonomy identification rank	GenBank	BOLD	
Number of query sequences	Number of species	Number of query sequences	Number of species	
90 and above	Order		15 (2)	5	21	
Family		13	
Subfamily	3		
Genus	9	15	
Species	9	5	
Below 90	Order		45 (2)	18	39	
Family	2	32	
Subfamily	29		
Genus	11	3	
Species	38	10	
Total	101	60	101	60	
Note:

The number in bracket () is the number of species shared in both pairwise identity percentage.

In general, a high proportion of these 101 top-hit search records were insufficiently identified, with only 46.5% of the GenBank and 14.9% of the BOLD records being identified to species rank. Besides that, only 21 records (15 species) and 38 records (21 species) from both GenBank and BOLD have pairwise identity percentage higher than 90% with the queried sequences. For instance, only nine out of the 21 GenBank records and five out of the 38 BOLD records have pairwise percentages higher than 90% with the queried sequences were identified to species rank.

Discussion

Species diversity among leaf beetles

Several earlier studies on species richness and biodiversity of Chrysomelidae from Sabah can be compared with the present study (Mohamedsaid, Salleh & Hassan, 1992; Mohamedsaid, 1995, 2004; Chung et al., 2000; Kishimoto-Yamada, Takizawa & Mahadimenakbar, 2016). For instance, Mohamedsaid, Salleh & Hassan (1992) reported 168 leaf beetle species from eight days sampling in Danum Valley Conservation Area, while Chung et al. (2000) reported 80 leaf beetle species from different habitat types in Sabah using different sampling approaches and Kishimoto-Yamada, Takizawa & Mahadimenakbar (2016) presented a checklist of 129 leaf beetle species from their 34 months sampling in Universiti Malaysia Sabah. The relatively low species richness recorded in this study is possibly caused by species poverty on the islands, different forest types sampled, different sampling methods and sampling efforts as in the previous studies (Wagner, 2000; Whittaker & Fernández-Palacios, 2007; Thormann, 2015).

The subfamily composition of the present study was also compared with the previous reports on Borneo and Oriental leaf beetle fauna (Kimoto, 1988; Mohamedsaid, 2004). The dominant leaf beetles on these islands are species of subfamilies Galerucinae (excluding Tribe Alticini) and Eumolpinae which is in accordance with the general trend throughout the Oriental region (Kimoto, 1988). Furthermore, the fact that Monolepta was the most speciose genus collected in this study also agrees with previous reports from Sabah and Borneo (Mohamedsaid, Salleh & Hassan, 1992; Mohamedsaid, 1995, 2004, 2006b; Kishimoto-Yamada, Takizawa & Mahadimenakbar, 2016). Although this study took samples from only 13 out of the 500 islands (∼3%) from Sabah, the number of species documented in the current study was ∼8.5% of the total number of reported species (803 species) in Borneo, which is a notable percentage in spite of the small proportion of sampled area in this study. This also suggests that species richness on the islands is possibly comparable to that of mainland habitats and that more species remain to be discovered.

This checklist also reveals the distribution of agricultural pest species on the islands, which is vital for the control of their dispersal. For example, Brontispa longissima, one of the most serious coconuts pests in the Pacific region, is commonly found on the sampled islands noted for human habitation, tourist activities and resorts. The sweet potatoes leaf beetle, Colasposoma auripenne was also found on an island with the cultivated Ipomoea batatas. It was possibly introduced to the island together with its host plant. Another agricultural pest species, Monolepta sp. 4 was found to have heavily defoliated the young shoots of a wide range of fruit trees (e.g. Citrus sp., Mangifera sp. and Sauropus androgynus).

DNA barcoding

Out of the 64 generated BINs, 60 unique BINs are new to BOLD and four non-unique BINs are existing records in BOLD, of which only one of the non-unique BINs, Scelodonta granulosa (BOLD:ADE7488) was previously recorded from Malaysia in BOLD (private data in BOLD, 2018) while the remaining 63 BINs are new to Malaysia. However, existing records of Scelodonta granulosa in BOLD were not made public and hence, this study resulted in the number of public records and BINs of Chrysomelidae from Malaysia in BOLD increasing to 174 records and 79 BINs, respectively. The high number of unique BINs and relatively low number of representatives from this region in BOLD highlight the urgency of studying the biodiversity in the region, in more depth.

All the non-unique BINs samples recorded in BOLD were collected from their known geographical distribution, except for Altica aenea. The known geographical region of Altica aenea does not include Pakistan as recorded in BOLD (Reid & Beatson, 2015) and possibly represents a new locality record for this species. However, re-examination of the specimens is necessary to validate this finding as one of the specimens recorded under the same BIN was identified as Altica birmanensis. The Brontispa longissima, one of the most serious invasive agricultural pests of the coconut palm, has a wide distribution across the Asia-Pacific region (Rethinam & Singh, 2007; CABI, 2018). The geographical distribution of the DNA barcoded Brontispa longissima specimens in BOLD were concordant with its known distribution, but none of these specimens were collected from Malaysia (BOLD, 2018; CABI, 2018). Therefore, the barcode compliant sequences generated in this study represent the first record of this species from Malaysia in BOLD.

The presence of cryptic species and polymorphic species was confirmed using the sequence analysis tools in BOLD. Six sequenced Nodina sp. specimens were revealed to be five different putative species OTUs as a result of the following analysis output combination: (1) assigned to five different BINs in BOLD, (2) intraspecific divergence (19.14%) higher than the distance to the nearest neighbour Basilepta sp. 1 (16.33%) in the Barcode Gap analysis, and (3) split into five different clusters in the Taxon ID Tree analysis (Fig. 2, highlighted in blue colour). However, these five OTUs are morphologically hard to distinguish from one another and thus, they are collectively treated as a single OTU (Kishimoto-Yamada, Takizawa & Mahadimenakbar, 2016) and excluded from the overall mean intraspecific and interspecific distance analyses.

Colour polymorphism is common in Chrysomelidae (Fujiyama, 1979; Verdyck et al., 1998) and many prominent cases have been reported (Mikhailov, 2001; Kurachi et al., 2002). In this study, 11 morphological identified OTUs were determined as four OTUs with colour polymorphism. Referring to the neighbour-joining tree (see Fig. 2), leaf beetles with colour polymorphism (highlighted in red colour) are Hoplosaenidea sp. 5, Monolepta sp. 5, Monolepta sp. 14 and genus indet. nr. Monolepta with mean intraspecific distances of 0%, 0.96%, 0.41%, and 0.2%, respectively. All the leaf beetle species that exhibited colour polymorphism in the study are from the subfamily Galerucinae and this is corroborated by previous reports (Maulik, 1936; Beenen, 2007; Prado, 2013). Colouration polymorphism is common within the genus Monolepta (Wagner, 2005) but probably the first record of it for genus Hoplosaenidea, even though Hoplosaenidea is well-known for the secondary sexual characteristics (Mohamedsaid & Furth, 2011). The colour polymorphism in leaf beetles is possibly caused by genetics, environment, host plant choices, seasonal and/or temperature variation during development (Petitpierre, 1988; Verdyck et al., 1998; Nahrung & Allen, 2005; Zverev et al., 2018).

Nonetheless, 97% of the sequences obtained from this study are new to GenBank. On top of that, out of the 21 queried sequences with pairwise identity percentage higher than 90%, only five were identified to species level in BLAST top-hits results. These are Brontispa longissima, Altica birmanensis and Altica engstroemi with pairwise identity percentages of 100%, 99.2% and 99.0%, respectively (see Table S1). However, both Altica birmanensis and Altica engstroemi were previously not recorded in Borneo and the latter species’ known distribution was only from northern Europe (Mohamedsaid, 2004; Reid & Beatson, 2015; GBIF, 2017). This has become complicated by the fact that the pairwise identity percentages of these two species sequences in GenBank is 99.5%, suggesting that they should be the same species, and that they were possibly misidentified as the locality of both specimens was recorded as being from Kerala, India. This conforms with previous reports on the poor quality of taxonomic identifications in GenBank (Bridge et al., 2003; Vilgalys, 2003; James Harris, 2003; Kristiansen et al., 2005). A similar situation was noted with for Altica aenea in BOLD, where one of the records under the same BIN was identified as Altica birmanensis. Re-examination of the specimen taxonomy identification confirmation is necessary to resolve this conflict. Besides that, the presence of other taxa in the top-hit search records and the high proportion of low pairwise identity percentages between queried sequences with GenBank and BOLD existing records highlights the poor representation and the urgency of this taxa from the region under study using molecular approaches.

Species Checklist

SUBFAMILY GALERUCINAE

Tribe ALTICINI Newman 1835

Genus Altica Geoffroy, 1762

Refer to Appendix A for the generic diagnosis of this genus extracted from respective key literature of the taxa.

Altica aenea (Olivier, 1808)

(Fig. 3A)

Examined materials (4). Pulau Tiga: BOR/COL 8071. Pulau Gaya: BOR/COL 8166, BOR/COL 8173, BOR/COL 9444.

Distribution in Sabah. Pulau Tiga, Pulau Gaya.

Distribution in the world. Altica aenea is widespread in tropical Australia and also found from India and Nepal east through Southeast Asia to the west Pacific Islands of Palau, Fiji, New Caledonia, Vanuatu and tropical Australia (Reid & Beatson, 2015).

Host plant. Ludwigia sp. (Reid & Beatson, 2015)

Barcode Index Number (BIN). BOLD:AAP8616

Intra-specific distance (%). Mean: 0  Max: 0

Remarks. BLAST top-hit result shows 99% similarity with Altica bermanensis and Altica engstromi. However, both species have not recorded in Sabah (Mohamedsaid, 2004; Reid & Beatson, 2015). So, the records in GenBank are probably misidentified.

Figure 3 Dorsal and ventral habitus of leaf beetle species.

(A) Altica aenea; (B) Aphthona sp. 1; (C) Argopistes sp. 1; (D) Argopistes sp. 2; (E) Erystus villicus; (F) Hemipyxis sp.; (G) Hyphasis sp.; (H) Lanka sp.

Genus Aphthona Chevrolat, 1837

Refer to Appendix A for the generic diagnosis of this genus extracted from respective key literature of the taxa.

Aphthona sp.

(Fig. 3B)

Examined materials (1). Pulau Mamutik: BOR/COL 9602.

Distribution in Sabah. Pulau Mamutik.

Barcode Index Number (BIN). BOLD:ADH3773

Remarks. Only found in Pulau Mamutik.

Genus Argopistes Motschulsky, 1860

Refer to Appendix A for the generic diagnosis of this genus extracted from respective key literature of the taxa. Two OTUs were identified and both OTUs were DNA barcoded.

Interspecific distance (%). Mean: 15.38  Max: 15.38

Argopistes sp. 1

(Fig. 3C)

Examined materials (11). Pulau Udar Kecil: BOR/COL 8442–8446, BOR/COL 9894–9895. Pulau Gaya: BOR/COL 9813, BOR/COL 9915–9917.

Distribution in Sabah. Pulau Udar Kecil, Pulau Gaya.

Barcode Index Number (BIN). BOLD:ADH5650

Remarks. Differentiated from Argopistes sp. 2 by black dorsum and yellow venter.

Argopistes sp. 2

(Fig. 3D)

Examined materials (2). Pulau Mamutik: BOR/COL 9608–9609.

Distribution in Sabah. Pulau Mamutik.

Barcode Index Number (BIN). BOLD:ADH5651

Remarks. Found near Citrus plant. Dorsal and ventral dark red in colour.

Genus Erystus Jacoby, 1885

Refer to Appendix A for the generic diagnosis of this genus extracted from respective key literature of the taxa.

Erystus villicus (Weise, 1892)

(Fig. 3E)

Examined materials (37). Pulau Gaya: BOR/COL 8134–8141, BOR/COL 8334–8341, BOR/COL 9332–9334, BOR/COL 9394–9395, BOR/COL 9400–9416.

Distribution in Sabah. Pulau Gaya.

Distribution in the world. Sabah, Philippines (Mohamedsaid, 2004).

Barcode Index Number (BIN). BOLD:ADH6322

Remarks. Usually found on Hibiscus tiliaceus near the beach with a great number of individuals. Heavily defoliate the host plant.

Genus Hemipyxis Chevrolat, 1836

Refer to Appendix A for the generic diagnosis of this genus extracted from respective key literature of the taxa.

Hemipyxis sp.

(Fig. 3F)

Examined materials (10). Pulau Gaya: BOR/COL 8187, BOR/COL 8213, BOR/COL 8236, BOR/COL 8325–8326, BOR/COL 9397, BOR/COL 9814–9815, BOR/COL 9924, BOR/COL 9961.

Distribution in Sabah. Pulau Gaya.

Barcode Index Number (BIN). N/A

Remarks. Only collected from Pulau Gaya. Body is yellow in colour.

Genus Hyphasis Harold, 1877

Refer to Appendix A for the generic diagnosis of this genus extracted from respective key literature of the taxa.

Hyphasis sp.

(Fig. 3G)

Examined materials (1). Pulau Dinawan: BOR/COL 8449.

Distribution in Sabah. Pulau Dinawan.

Barcode Index Number (BIN). BOLD:ADH5610

Remarks. Only found in Pulau Dinawan, near to deforested area.

Genus LankaMaulik, 1926

Refer to Appendix A for the generic diagnosis of this genus extracted from respective key literature of the taxa.

Lanka sp.

(Fig. 3H)

Examined materials (1). Pulau Gaya: BOR/COL 8097.

Distribution in Sabah. Pulau Gaya.

Barcode Index Number (BIN). BOLD:ADH7255

Remarks. Collected from a plant near to the river in Pulau Gaya.

Genus Schenklingia Csiki & Heikertinger, 1940

Refer to Appendix A for the generic diagnosis of this genus extracted from respective key literature of the taxa.

Schenklingia sp.

(Fig. 4A)

Examined materials (1). Pulau Gaya: BOR/COL 9429.

Distribution in Sabah. Pulau Gaya.

Barcode Index Number (BIN). BOLD:ADH3903

Remarks. Body colour dark red, first three and 11th antennal segment orange–brown colour and remaining antennal segments black in colour.

Figure 4 Dorsal and ventral habitus of leaf beetle species.

(A) Schenklingia sp.; (B) Aulacophora sp.; (C) Strobiderus sp.; (D) Hoplosaenidea malayensis; (E) Hoplosaenidea sp. 1; (F) Hoplosaenidea sp. 2; (G) Hoplosaenidea sp. 3; (H) Hoplosaenidea sp. 4.

Tribe LUPERINI Leng 1920

Subtribe AULACOPHORINA Wilcox 1972

Section Aulacophorites Chapius 1875

Genus Aulacophora Dejean, 1835

Refer to Appendix A for the generic diagnosis of this genus extracted from respective key literature of the taxa.

Aulacophora sp.

(Fig. 4B)

Examined materials (7). Pulau Gaya: BOR/COL 8103, BOR/COL 8184, BOR/COL 8321, BOR/COL 8331, BOR/COL 9462–9464.

Distribution in Sabah. Pulau Gaya.

Barcode Index Number (BIN). BOLD:ADH4212

Remarks. Found on the plants near river area in Pulau Gaya.

Subtribe LUPERINA Wilcox 1965

Section Doryscites Wilcox 1973

Genus StrobiderusJacoby, 1884

Refer to Appendix A for the generic diagnosis of this genus extracted from respective key literature of the taxa.

Strobiderus sp.

(Fig. 4C)

Examined materials (7). Pulau Tiga: BOR/COL 6995–6999, BOR/COL 9155, BOR/COL 9156.

Distribution in Sabah. Pulau Tiga.

Barcode Index Number (BIN). BOLD:ADH6702

Remarks. Collected from the ventral part of the leaves of plant family Araceae.

Section Luperites Chapius 1875

Genus HoplosaenideaLaboissiere, 1933

Refer to Appendix A for the generic diagnosis of this genus extracted from respective key literature of the taxa. Seven OTUs were identified and six OTUs were DNA barcoded.

Interspecific distance (%). Mean: 17.70  Max: 22.35

Hoplosaenidea malayensis (Jacoby, 1884)

(Fig. 4D)

Examined materials (14). Pulau Gaya: BOR/COL 8188–8193, BOR/COL 8330, BOR/COL 9854–9856. Pulau Sapangar: BOR/COL 8425, BOR/COL 9680–9681. Pulau Udar Besar: BOR/COL 8440.

Distribution in Sabah. Pulau Gaya, Pulau Sapangar, Pulau Udar Besar.

Distribution in the world. Peninsular Malaysia, Sabah, Sarawak, Sumatra (Mohamedsaid, 2004).

Host plant. Barringtonia sp. (Lecythidaceae) (Mohamedsaid, 2004)

Barcode Index Number (BIN). BOLD:ADH4031

Intraspecific distance (%). Mean: 0.1  Max: 0.15

Remarks. Whole body yellow in colour, usually found in few of individuals on a single plant.

Hoplosaenidea sp. 1

(Fig. 4E)

Examined materials (1). Pulau Tiga: BOR/COL 7000.

Distribution in Sabah. Pulau Tiga.

Barcode Index Number (BIN). BOLD:ADH3897

Remarks. Body completely creamy white in colour.

Hoplosaenidea sp. 2

(Fig. 4F)

Examined materials (6). Pulau Tiga: BOR/COL 8538. Pulau Gaya: BOR/COL 9430–9434.

Distribution in Sabah. Pulau Tiga, Pulau Gaya.

Barcode Index Number (BIN). BOLD:ADH4030

Remarks. Whole body banana yellow in colour, and elytra with two longitudinally black stripes.

Hoplosaenidea sp. 3

(Fig. 4G)

Examined materials (1). Pulau Gaya: BOR/COL 8268.

Distribution in Sabah. Pulau Gaya.

Barcode Index Number (BIN). N/A

Remarks. Whole body red–orange in colour.

Hoplosaenidea sp. 4

(Fig. 4H)

Examined materials (1). Pulau Gaya: BOR/COL 8095.

Distribution in Sabah. Pulau Gaya.

Barcode Index Number (BIN). BOLD:ADH4029

Remarks. Similar to Hoplosaenidea sp. 6, different by thorax and elytra colouration, and the 9th antennae segment on basal half white and on apical half black.

Hoplosaenidea sp. 5

(Figs. 5A–5B)

Examined materials (2). Pulau Mantukod: BOR/COL 9720–9721.

Distribution in Sabah. Pulau Mantukod.

Barcode Index Number (BIN). BOLD:ADH4033

Intraspecific distance (%). Mean: 0  Max: 0

Remarks. Phenotypic polymorphism, with difference in size and body colour.

Figure 5 Dorsal and ventral habitus of leaf beetle species.

(A and B) Hoplosaenidea sp. 5; (C) Hoplosaenidea variabilis; (D) Metrioidea grandis; (E) Monolepta sp. 1; (F) Monolepta sp. 2; (G) Monolepta sp. 3; (H) Monolepta sp. 4.

Hoplosaenidea variabilis (Jacoby, 1894)

(Fig. 5C)

Examined materials (1). Pulau Udar Besar: BOR/COL 9638.

Distribution in Sabah. Pulau Udar Besar.

Distribution in the world. Peninsular Malaysia, Sarawak, Java (Mohamedsaid, 2004).

Barcode Index Number (BIN). BOLD:ADH4032

Remarks. Head and thorax maroon red colour, and elytra with metallic bluish-green colour.

Section MONOLEPTITES Chapius 1875

Genus MetrioideaFairmaire, 1881

Refer to Appendix A for the generic diagnosis of this genus extracted from respective key literature of the taxa.

Metrioidea grandis (Allard, 1889)

(Fig. 5D)

Examined materials (55). Pulau Gaya: BOR/COL 8094, BOR/COL 8106–8108, BOR/COL 8121–8122, BOR/COL 8131–8132, BOR/COL 8171–8172, BOR/COL 8238, BOR/COL 8241–8245, BOR/COL 8270–8273, BOR/COL 8283–8304, BOR/COL 8310, BOR/COL 9428, BOR/COL 9465–9467, BOR/COL 9480, BOR/COL 9494. Pulau Sapangar: BOR/COL 8417, BOR/COL 8429–8433.

Distribution in Sabah. Pulau Gaya, Pulau Sapangar.

Distribution in the world. Sarawak, Sabah, Borneo (Mohamedsaid, 2004).

Barcode Index Number (BIN). BOLD:ADH7177

Intraspecific distance (%). Mean: 1.99  Max: 1.99

Remarks. Body orange in colour. Elytra become semi-transparent after soaking in ethanol.

Genus Monolepta Erichson, 1843

Refer to Appendix A for the generic diagnosis of this genus extracted from respective key literature of the taxa. 18 OTUs were identified and 16 OTUs were DNA barcoded.

Interspecific distance (%). Mean: 15.88  Max: 24.60

Monolepta sp. 1

(Fig. 5E)

Examined materials (53). Pulau Gaya: BOR/COL 8209–8211, BOR/COL 8235, BOR/COL 8250, BOR/COL 8323, BOR/COL 8332, BOR/COL 9396, BOR/COL 9460, BOR/COL 9805, BOR/COL 9836–9840, BOR/COL 9932. Pulau Sapangar: BOR/COL 8427, BOR/COL 8435–8436, BOR/COL 9672, BOR/COL 9674, BOR/COL 9690, BOR/COL 9699, BOR/COL 9709–9711. Pulau Udar Besar: BOR/COL 8441. Pulau Sapi: BOR/COL 9215–9220, BOR/COL 9243. Pulau Sulug: BOR/COL 9619–9622, BOR/COL 9624–9633, BOR/COL 9635–9636. Pulau Mantukod: BOR/COL 9736. Pulau Udar Kecil: BOR/COL 9899–9900.

Distribution in Sabah. Pulau Gaya, Pulau Sapangar, Pulau Udar Besar, Pulau Sapi, Pulau Sulug, Pulau Mantukod, Pulau Udar Kecil.

Barcode Index Number (BIN). BOLD:ADH4138

Intraspecific distance (%). Mean: 0.92  Max: 0.92

Remarks. Whole body yellow in colour with the brown or orange tibia.

Monolepta sp. 2

(Fig. 5F)

Examined materials (4). Pulau Gaya: BOR/COL 6931, BOR/COL 9442. Pulau Tiga: BOR/COL 8526, BOR/COL 9774.

Distribution in Sabah. Pulau Gaya, Pulau Tiga.

Barcode Index Number (BIN). BOLD:ADH7139

Remarks. Body length around 2–3 mm. Black colour elytra with two distinct white bands. Last ventrite segment black.

Monolepta sp. 3

(Fig. 5G)

Examined materials (8). Pulau Gaya: BOR/COL 6924–6926, BOR/COL 9423, BOR/COL 9468–9470. Pulau Tiga: BOR/COL 9778.

Distribution in Sabah. Pulau Gaya, Pulau Tiga.

Barcode Index Number (BIN). BOLD:ADH4196

Remarks. This species especially abundant during the flowering season, with deep red colour head, thorax and abdomen, and black colour elytra, last antennae segment black in colour.

Monolepta sp. 4

(Fig. 5H)

Examined materials (174). Pulau Gaya: BOR/COL 6921–6923, BOR/COL 9335–9392, BOR/COL 9471–9478, BOR/COL 9821. Pulau Dinawan: BOR/COL 8455, BOR/COL 8492–8505, BOR/COL 9755. Pulau Mengalum: BOR/COL 9249, BOR/COL 9967–9976. Pulau Manukan: BOR/COL 9558–9579. Pulau Udar Besar: BOR/COL 9640–9667. Pulau Peduk: BOR/COL 9860–9872. Pulau Udar Kecil: BOR/COL 9901–9914.

Distribution in Sabah. Pulau Gaya, Pulau Dinawan, Pulau Mengalum, Pulau Manukan, Pulau Udar Besar, Pulau Peduk, Pulau Udar Kecil.

Barcode Index Number (BIN). BOLD:ADH6840

Remarks. Heavily defoliate Citrus sp., Mangifera sp. and Sauropus androgynus plants young shoots.

Monolepta sp. 5

(Figs. 6A–6D)

Examined materials (20). Pulau Gaya: BOR/COL 8178, BOR/COL 8180, BOR/COL 8251. Pulau Sapi: BOR/COL 8348. Pulau Manukan: BOR/COL 8403, BOR/COL 9583–9585, BOR/COL 9592. Pulau Mamutik: BOR/COL 9603. Pulau Mantukod: BOR/COL 9718–9719, BOR/COL 9722–9723, BOR/COL 9733–9735. Pulau Dinawan: BOR/COL 9740–9741. Pulau Udar Kecil: BOR/COL 9897.

Distribution in Sabah. Pulau Gaya, Pulau Sapi, Pulau Manukan, Pulau Mamutik, Pulau Mantukod, Pulau Dinawan, Pulau Udar Kecil.

Barcode Index Number (BIN). BOLD:ADH4050

Intraspecific distance (%). Mean: 0.96  Max: 1.69

Remarks. This species exhibits phenotypic polymorphism, with four different phenotypic characters, one fully milky white in colour, one with suture and elytra edge black in colour, one elytra with two dark brown bands separated by light brown bands, and one elytra with two dark brown bands interconnected by dark brown suture but separated by two light brown bands.

Figure 6 Dorsal and ventral habitus of leaf beetle species.

(A–D) Monolepta sp. 5; (E) Monolepta sp. 6; (F) Monolepta sp. 7; (G) Monolepta sp. 8; (H) Monolepta sp. 9.

Monolepta sp. 6

(Fig. 6E)

Examined materials (1). Pulau Tiga: BOR/COL 8531.

Distribution in Sabah. Pulau Tiga.

Barcode Index Number (BIN). BOLD:ADH6249

Remarks. Found resting on the beach on Ipomoea species. Head and elytra deep red in colour with thorax creamy white in colour.

Monolepta sp. 7

(Fig. 6F)

Examined materials (6). Pulau Sapangar: BOR/COL 8426, BOR/COL 8437–8439, BOR/COL 9677, BOR/COL 9717.

Distribution in Sabah. Pulau Sapangar.

Barcode Index Number (BIN). BOLD:ADH4051

Remarks. Usually found on the highest point in Pulau Sapangar.

Monolepta sp. 8

(Fig. 6G)

Examined materials (22). Pulau Gaya: BOR/COL 8314, BOR/COL 9299–9301, BOR/COL 9824, BOR/COL 9826–9835, BOR/COL 9841, BOR/COL 9939–9944.

Distribution in Sabah. Pulau Gaya.

Barcode Index Number (BIN). BOLD:ADH7150

Remarks. Only collected from Pulau Gaya, light yellow in colour.

Monolepta sp. 9

(Fig. 6H)

Examined materials (15). Pulau Gaya: BOR/COL 8276, BOR/COL 9330, BOR/COL 9435–9437, BOR/COL 9440–9441, BOR/COL 9443, BOR/COL 9825. Pulau Tiga: BOR/COL 8525, BOR/COL 9153, BOR/COL 9165–9166, BOR/COL 9779. Pulau Sapangar: BOR/COL 9684.

Distribution in Sabah. Pulau Gaya, Pulau Tiga, Pulau Sapangar.

Barcode Index Number (BIN). BOLD:ADH7149

Intraspecific distance (%). Mean: 1.32  Max: 1.83

Remarks. Black colour head with milky white colour thorax and black colour elytra with a white band in the middle of the elytra.

Monolepta sp. 10

(Fig. 7A)

Examined materials (3). Pulau Gaya: BOR/COL 8104, BOR/COL 8181. Pulau Mantukod: BOR/COL 9732.

Distribution in Sabah. Pulau Gaya, Pulau Mantukod.

Barcode Index Number (BIN). BOLD:ADH7148

Intraspecific distance (%). Mean: 0  Max: 0

Remarks. Orange colour head and thorax, semi-transparent elytra with light green abdomen.

Figure 7 Dorsal and ventral habitus of leaf beetle species.

(A) Monolepta sp. 10; (B) Monolepta sp. 11; (C) Monolepta sp. 12; (D) Monolepta sp. 13; (E and F) Monolepta sp. 14; (G) Monolepta sp. 15; (H) Monolepta sp. 16.

Monolepta sp. 11

(Fig. 7B)

Examined materials (1). Pulau Gaya: BOR/COL 8119.

Distribution in Sabah. Pulau Gaya.

Barcode Index Number (BIN). BOLD:ADH7140

Remarks. Similar to Monolepta sp. 18, with the difference on the elytra patterns.

Monolepta sp. 12

(Fig. 7C)

Examined materials (1). Pulau Tiga: BOR/COL 9201.

Distribution in Sabah. Pulau Tiga.

Barcode Index Number (BIN). BOLD:ADH4195

Remarks. Collected from random sweeping along the Pagong-Pagong trail in Pulau Tiga.

Monolepta sp. 13

(Fig. 7D)

Examined materials (1). Pulau Sapangar: BOR/COL 9678.

Distribution in Sabah. Pulau Sapangar.

Barcode Index Number (BIN). N/A

Remarks. Body length 2–3 mm. Only collected from Pulau Sapangar.

Monolepta sp. 14

(Figs. 7E–7F)

Examined materials (9). Pulau Sapi: BOR/COL 9223, BOR/COL 9240. Pulau Gaya: BOR/COL 9418–9419, BOR/COL 9450, BOR/COL 9842. Pulau Manukan: BOR/COL 9556–9557. Pulau Tiga: BOR/COL 9766.

Distribution in Sabah. Pulau Sapi, Pulau Gaya, Pulau Manukan, Pulau Tiga.

Barcode Index Number (BIN). BOLD:ADH4966

Intraspecific distance (%). Mean: 0.41  Max: 0.62

Remarks. Possibly exhibits sexual dimorphism.

Monolepta sp. 15

(Fig. 7G)

Examined materials (7). Pulau Dinawan: BOR/COL 8456. Pulau Gaya: BOR/COL 9438–9439. Pulau Mamutik: BOR/COL 9600. Pulau Udar Besar: BOR/COL 9639, BOR/COL 9670. Pulau Peduk: BOR/COL 9883.

Distribution in Sabah. Pulau Dinawan, Pulau Gaya, Pulau Mamutik, Pulau Udar Besar, Pulau Peduk.

Barcode Index Number (BIN). BOLD:ADH4198

Remarks. Black colour head with the yellow thorax, black elytra with one yellow band in the middle.

Monolepta sp. 16

(Fig. 7H)

Examined materials (3). Pulau Gaya: BOR/COL 9424, BOR/COL 9445, BOR/COL 9958.

Distribution in Sabah. Pulau Gaya.

Barcode Index Number (BIN). N/A

Remarks. Whole body brown in colour, only found in Pulau Gaya.

Monolepta sp. 17

(Fig. 8A)

Examined materials (1). Pulau Gaya: BOR/COL 9449.

Distribution in Sabah. Pulau Gaya.

Barcode Index Number (BIN). BOLD:ADH7141

Remarks. Whole body white in colour, meso- and metasternum light brown in colour.

Figure 8 Dorsal and ventral habitus of leaf beetle species.

(A) Monolepta sp. 17; (B) Monolepta sp. 18; (C) Ochralea nigripes; (D) Clitena sp.; (E) Sumatrasia sp.; (F) Dercetina sp.; (G and H) genus indet. nr. Monolepta.

Monolepta sp. 18

(Fig. 8B)

Examined materials (1). Pulau Sapangar: BOR/COL 9679.

Distribution in Sabah. Pulau Sapangar.

Barcode Index Number (BIN). BOLD:ADH4197

Remarks. Differentiate from Monolepta sp. 11 by the dark colour patterns on the elytra.

Genus OchraleaClark, 1865

Refer to Appendix A for the generic diagnosis of this genus extracted from respective key literature of the taxa.

Ochralea nigripes (Olivier, 1808)

(Fig. 8C)

Examined materials (130). Pulau Tiga: BOR/COL 8002–8005, BOR/COL 8015, BOR/COL 8017–8039, BOR/COL 8048–8050, BOR/COL 8515–8518, BOR/COL 8520–8524, BOR/COL 8528–8530, BOR/COL 8542–8544, BOR/COL 8555–8558, BOR/COL 9121–9147, BOR/COL 9160–9164, BOR/COL 9198–9199, BOR/COL 9202–9206, BOR/COL 9780–9789, BOR/COL 9798–9799. Pulau Gaya: BOR/COL 8201, BOR/COL 8212, BOR/COL 8234, BOR/COL 8319–8320, BOR/COL 8322, BOR/COL 8327, BOR/COL 9461, BOR/COL 9956. Pulau Sapi: BOR/COL 8356, BOR/COL 8362. Pulau Mamutik: BOR/COL 8408, BOR/COL 8411–8416, BOR/COL 9604–9607. Pulau Udar Besar: BOR/COL 9668–9669. Pulau Sapangar: BOR/COL 9692–9696, BOR/COL 9707.

Distribution in Sabah. Pulau Tiga, Pulau Gaya, Pulau Sapi, Pulau Mamutik, Pulau Udar Besar, Pulau Sapangar.

Distribution in the world. This species is widely distributed in many east Asian countries, such as Malaysia, Indonesia, Philippines, Cambodia, Burma, Myanmar, Thailand, Brunei, Laos, Vietnam, Singapore, northwards to Southern China, westwards to Bangladesh, eastwards to Sulawesi and as far as the Wallace-line (Hazmi & Wagner, 2010).

Barcode Index Number (BIN). BOLD:ADH4213

Intraspecific distance (%). Mean: 0.7  Max: 1.06

Remarks. 8–10 mm body length, with colour variations of yellow and yellow–orange body colour. Very abundant especially in Pulau Gaya and Pulau Tiga. Few individuals collected in between leaf litters and twigs from the ground.

Tribe GALERUCINI Laboissiere 1921

Section Coelomerites Chapius 1875

Genus Clitena Baly, 1864

Refer to Appendix A for the generic diagnosis of this genus extracted from respective key literature of the taxa.

Clitena sp.

(Fig. 8D)

Examined materials (2). Pulau Manukan: BOR/COL 8399, BOR/COL 9580.

Distribution in Sabah. Pulau Manukan.

Barcode Index Number (BIN). BOLD:ADH3702

Remarks. Found near the sunset point shelter in Pulau Manukan.

Tribe METACYCLINI Leng 1920

Genus SumatrasiaJacoby, 1884

Refer to Appendix A for the generic diagnosis of this genus extracted from respective key literature of the taxa.

Sumatrasia sp.

(Fig. 8E)

Examined materials (1). Pulau Sapi: BOR/COL 6938.

Distribution in Sabah. Pulau Sapi.

Barcode Index Number (BIN). BOLD:ADH4430

Remarks. Whole body yellow in colour. Collected along the trail in Pulau Sapi.

Tribe SERMYLINI Wilcox 1965

Section Antiphites Chapius 1875

Genus DercetinaGressitt & Kimoto, 1963

Refer to Appendix A for the generic diagnosis of this genus extracted from respective key literature of the taxa.

Dercetina sp.

(Fig. 8F)

Examined materials (5). Pulau Gaya: BOR/COL 8150. Pulau Sapangar: BOR/COL 8428, BOR/COL 8434, BOR/COL 9671, BOR/COL 9713.

Distribution in Sabah. Pulau Gaya, Pulau Sapangar.

Barcode Index Number (BIN). BOLD:ADH3896

Intraspecific distance (%). Mean: 0.3  Max: 0.3

Remarks. Head, thorax and the basal half of elytra red colour, and apical half black. Last ventrite visible from dorsal.

Genus indet. nr. Monolepta

(Figs. 8G–8H and 9A)

Examined materials (14). Pulau Gaya: BOR/COL 8277. Pulau Peduk: BOR/COL 9873–9877, BOR/COL 9884–9885, BOR/COL 9888–9893.

Distribution in Sabah. Pulau Gaya, Pulau Peduk.

Barcode Index Number (BIN). BOLD:ADH3996

Intraspecific distance (%). Mean: 0.2  Max: 0.3

Remarks. Possibly exhibit phenotypic polymorphism with three different patterns and colourations on the elytra. These three patterns also observed at UMS hill based on second author collection.

Figure 9 Dorsal and ventral habitus of leaf beetle species.

(A) Genus indet. nr. Monolepta; (B) Scelodonta sp.; (C) Colasposoma auripenne; (D) Aulacia sp.; (E) Colaspoides sp. 1; (F) Colaspoides tuberculata; (G) Basilepta sp. 1; (H) Basilepta sp. 2.

SUBFAMILY EUMOLPINAE

Tribe ADOXINI Jacoby, 1908

Section Scelodontites Chapius 1874

Genus Scelodonta Westwood, 1837

Refer to Appendix A for the generic diagnosis of this genus extracted from respective key literature of the taxa.

Scelodonta granulosaBaly, 1867

(Fig. 9B)

Examined materials (2). Pulau Mengalum: BOR/COL 9531. Pulau Sapangar: BOR/COL 9682.

Distribution in Sabah. Pulau Mengalum, Pulau Sapangar.

Distribution in the world. Borneo, China, India, Laos, Sarawak, Sulawesi, Thailand, Vietnam (Mohamedsaid, 2009).

Barcode Index Number (BIN). BOLD:ADE7488

Remarks. Iridescent body colour with the red colour tibia.

Tribe COLASPOSOMINI Springlova 1960

Section Colasposomites Wilcox 1982

Genus Colasposoma Laporte, 1833

Refer to Appendix A for the generic diagnosis of this genus extracted from respective key literature of the taxa.

Colasposoma auripenne Motschulsky, 1860

(Fig. 9C)

Examined materials (2). Pulau Dinawan: BOR/COL 9753–9754.

Distribution in Sabah. Pulau Dinawan.

Distribution in the world. Borneo, Cambodia, China, India, Japan, Java, Laos. Myanmar, Peninsular Malaysia, Taiwan, Thailand, Vietnam (Mohamedsaid, 2004).

Host plant. Ipomoea aquatica, Ipomoea batatas, Ipomoea fistulosa, Ipomoea hispida, Ipomoea indica, Ipomoea palmate, Ipomoea pestigridis and Ipomoea pilosa (Kalaichelvan & Verma, 2005).

Barcode Index Number (BIN). BOLD:ADH6210

Remarks. This species was found on the cultivated sweet potatoes plant, Ipomoea batatas.

Tribe EUMOLPINI Jacoby, 1908

Section Endocephalites Chapius 1874

Genus AulaciaBaly, 1867

Refer to Appendix A for the generic diagnosis of this genus extracted from respective key literature of the taxa.

Aulacia sp.

(Fig. 9D)

Examined materials (2). Pulau Tiga: BOR/COL 9154, BOR/COL 9200.

Distribution in Sabah. Pulau Tiga.

Barcode Index Number (BIN). N/A

Remarks. Dark brown in colour. Found from Pagong-Pagong trail, Pulau Tiga.

Genus Colaspoides Laporte, 1833

Refer to Appendix A for the generic diagnosis of this genus extracted from respective key literature of the taxa. 2 OTUs were identified and both OTUs were DNA barcoded.

Interspecific distance (%). Mean: 23.03  Max: 23.03

Colaspoides sp. 1

(Fig. 9E)

Examined materials (9). Pulau Tiga: BOR/COL 8545–8546. Pulau Gaya: BOR/COL 9398, BOR/COL 9454–9458. Pulau Sapangar: BOR/COL 9691.

Distribution in Sabah. Pulau Tiga, Pulau Gaya, Pulau Sapangar.

Barcode Index Number (BIN). BOLD:ADH4442

Remarks. 1st–8th antennae segments yellow–brown, 9th–11th antennae segments black, dorsum and leg yellow–brown.

Colaspoides tuberculataBaly, 1867

(Fig. 9F)

Examined materials (1). Pulau Gaya: BOR/COL 9858.

Distribution in Sabah. Pulau Gaya.

Distribution in the world. Sabah, Sarawak (Mohamedsaid, 2004; Kishimoto-Yamada, Takizawa & Mahadimenakbar, 2016).

Barcode Index Number (BIN). BOLD:ADH4443

Remarks. Antennae black, body colour iridescent colour.

Tribe NODININI Chen 1940

Section Nodostomini Jacoby, 1908

Genus BasileptaBaly, 1860

Refer to Appendix A for the generic diagnosis of this genus extracted from respective key literature of the taxa. Six OTUs were identified and four OTUs were DNA barcoded.

Interspecific distance (%). Mean: 18.02  Max: 21.63

Basilepta sp. 1

(Fig. 9G)

Examined materials (21). Pulau Gaya: BOR/COL 8202, BOR/COL 8237, BOR/COL 9296, BOR/COL 9302–9310, BOR/COL 9843–9848, BOR/COL 9918–9919. Pulau Mantukod: BOR/COL 9737.

Distribution in Sabah. Pulau Gaya, Pulau Mantukod.

Barcode Index Number (BIN). BOLD:ADH5567

Remarks. Pronotum strongly punctate, body dark brown in colour.

Basilepta sp. 2

(Fig. 9H)

Examined materials (9). Pulau Gaya: BOR/COL 6930, BOR/COL 9311–9316, BOR/COL 9819. Pulau Sapangar: BOR/COL 9683.

Distribution in Sabah. Pulau Gaya, Pulau Sapangar.

Barcode Index Number (BIN). BOLD:ADH5568

Remarks. Thorax strongly punctate, elytra weakly punctate than pronotum.

Basilepta sp. 3

(Fig. 10A)

Examined materials (6). Pulau Tiga: BOR/COL 9158, BOR/COL 9170, BOR/COL 9187, BOR/COL 9757, BOR/COL 9769. Pulau Sapangar: BOR/COL 9714.

Distribution in Sabah. Pulau Tiga, Pulau Sapangar.

Barcode Index Number (BIN). BOLD:ADI3390

Remarks. Pronotum impunctate, elytra not strongly punctate.

Figure 10 Dorsal and ventral habitus of leaf beetle species.

(A) Basilepta sp. 3; (B) Basilepta sp. 4; (C) Basilepta sp. 5; (D) Basilepta sp. 6; (E–H) Nodina sp.

Basilepta sp. 4

(Fig. 10B)

Examined materials (10). Pulau Tiga: BOR/COL 8064–8065, BOR/COL 9758–9765.

Distribution in Sabah. Pulau Tiga.

Barcode Index Number (BIN). BOLD:ADH4287

Remarks. Body red–brown in colour. Pronotum weakly punctate than on elytra.

Basilepta sp. 5

(Fig. 10C)

Examined materials (13). Pulau Gaya: BOR/COL 8102, BOR/COL 8198, BOR/COL 8324, BOR/COL 9331, BOR/COL 9420, BOR/COL 9822–9823, BOR/COL 9927–9931. Pulau Sapangar: BOR/COL 9686.

Distribution in Sabah. Pulau Gaya, Pulau Sapangar.

Barcode Index Number (BIN). N/A

Remarks. Pronotum impunctate on median area, strongly punctate laterally.

Basilepta sp. 6

(Fig. 10D)

Examined materials (1). Pulau Mengalum: BOR/COL 9977.

Distribution in Sabah. Pulau Mengalum.

Barcode Index Number (BIN). N/A

Remarks. Only collected from Pulau Mengalum.

Genus Nodina Motschulsky, 1858

Refer to Appendix A for the generic diagnosis of this genus extracted from respective key literature of the taxa.

Nodina sp.

(Figs. 10E–10H and 11A)

Examined materials (159). Pulau Tiga: BOR/COL 6994, BOR/COL 8047, BOR/COL 8527, BOR/COL 8547–8548, BOR/COL 8559, BOR/COL 9148–9152, BOR/COL 9167, BOR/COL 9171–9186, BOR/COL 9188–9197, BOR/COL 9210–9211, BOR/COL 9768, BOR/COL 9782, BOR/COL 9790–9797, BOR/COL 9800–9802. Pulau Gaya: BOR/COL 8179, BOR/COL 8197, BOR/COL 8240, BOR/COL 8258, BOR/COL 8328, BOR/COL 9297, BOR/COL 9298, BOR/COL 9319–9422, BOR/COL 9806–9811, BOR/COL 9920–9923, BOR/COL 9933–9935, BOR/COL 9945, BOR/COL 9955, BOR/COL 9959. Pulau Sapi: BOR/COL 8350–8352, BOR/COL 8365–8366, BOR/COL 8369–8373, BOR/COL 9221–9222, BOR/COL 9224–9237, BOR/COL 9241. Pulau Manukan: BOR/COL 8398, BOR/COL 8400–8402, BOR/COL 9552, BOR/COL 9555, BOR/COL 9590. Pulau Sapangar: BOR/COL 8418–8421, BOR/COL 9676, BOR/COL 9689, BOR/COL 9701–9706, BOR/COL 9712. Pulau Dinawan: BOR/COL 8447–8448, BOR/COL 8460–8469, BOR/COL 8473–8485, BOR/COL 9738–9739, BOR/COL 9748–9749. Pulau Mantukod: BOR/COL 8506, BOR/COL 8510, BOR/COL 9731.

Distribution in Sabah. Pulau Tiga, Pulau Gaya, Pulau Sapi, Pulau Manukan, Pulau Sapangar, Pulau Dinawan, Pulau Mantukod.

Barcode Index Number (BIN). BOLD:ADI2797, BOLD:ADI2798, BOLD:ADI3779, BOLD:ADI3780, BOLD:ADI3781

Remarks. These species are so closely similar on outer morphological traits that we refrain from sorting them into morphological species at present. Six random individuals selected for sequencing results in five different OTUs.

Section Pagriites Lefevre 1885

Genus PagriaLefevre, 1884

Refer to Appendix A for the generic diagnosis of this genus extracted from respective key literature of the taxa.

Pagria sp.

(Fig. 11B)

Examined materials (1). Pulau Gaya: BOR/COL 9479.

Distribution in Sabah. Pulau Gaya.

Barcode Index Number (BIN). BOLD:ACW8270

Remarks. Head and thorax black in colour and elytra brown in colour.

Figure 11 Dorsal and ventral habitus of leaf beetle species.

(A) Nodina sp.; (B) Pagria sp.; (C) Rhyparida sp. 1; (D) Rhyparida sp. 2; (E) Cleorina malayana; (F) Brontispa longissima; (G) Gonophora sp.; (H) Dactylispa sp. 1.

Section Metachromites Chapius 1874

Genus Rhyparida Baly, 1861

Refer to Appendix A for the generic diagnosis of this genus extracted from respective key literature of the taxa. Two OTUs were identified and both OTUs were DNA barcoded.

Interspecific distance (%). Mean: 16.91  Max: 17.79

Rhyparida sp. 1

(Fig. 11C)

Examined materials (74). Pulau Sapi: BOR/COL 6939–6940, BOR/COL 8355, BOR/COL 9238–9239, BOR/COL 9242. Pulau Tiga: BOR/COL 8063, BOR/COL 9159, BOR/COL 9168, BOR/COL 9767. Pulau Sapangar: BOR/COL 8422, BOR/COL 9675, BOR/COL 9697–9698, BOR/COL 9700. Pulau Dinawan: BOR/COL 8450–8451, BOR/COL 8470, BOR/COL 9742–9746. Pulau Mantukod: BOR/COL 8508, BOR/COL 9724–9729. Pulau Mengalum: BOR/COL 9252–9254, BOR/COL 9258–9265, BOR/COL 9268–9270, BOR/COL 9459, BOR/COL 9496–9501, BOR/COL 9518–9530, BOR/COL 9532–9537, BOR/COL 9540, BOR/COL 9549. Pulau Gaya: BOR/COL 9329, BOR/COL 9425.

Distribution in Sabah. Pulau Sapi, Pulau Tiga, Pulau Sapangar, Pulau Dinawan, Pulau Mantukod, Pulau Mengalum, Pulau Gaya.

Barcode Index Number (BIN). BOLD:ADH5562

Intraspecific distance (%). Mean: 0.91  Max: 1.53

Remarks. Anterior femora with or without weak spine.

Rhyparida sp. 2

(Fig. 11D)

Examined materials (64). Pulau Gaya: BOR/COL 8196, BOR/COL 8262, BOR/COL 8269, BOR/COL 8311, BOR/COL 9322–9328, BOR/COL 9414, BOR/COL 9426–9427. Pulau Dinawan: BOR/COL 8471–8472. Pulau Sapi: BOR/COL 9212–9214. Pulau Mengalum: BOR/COL 9250–9251, BOR/COL 9255–9257, BOR/COL 9266–9267, BOR/COL 9502–9506, BOR/COL 9509–9517, BOR/COL 9538–9539, BOR/COL 9541–9548, BOR/COL 9963–9966, BOR/COL 9978–9980. Pulau Manukan: BOR/COL 9581. Pulau Sapangar: BOR/COL 9708, BOR/COL 9715–9716. Pulau Mantukod: BOR/COL 9730. Pulau Udar Kecil: BOR/COL 9896, BOR/COL 9898.

Distribution in Sabah. Pulau Gaya, Pulau Dinawan, Pulau Sapi, Pulau Mengalum, Pulau Manukan, Pulau Sapangar, Pulau Mantukod, Pulau Udar Kecil.

Barcode Index Number (BIN). BOLD:ADH5563

Remarks. Anterior femore with well-developed spine on inner margin.

Section Typophorites Chapius 1874

Genus Cleorina Lefevre, 1885

Refer to Appendix A for the generic diagnosis of this genus extracted from respective key literature of the taxa.

Cleorina malayana (Jacoby, 1896)

(Fig. 11E)

Examined materials (8). Pulau Manukan: BOR/COL 9593–9599. Pulau Sulug: BOR/COL 9637.

Distribution in Sabah. Pulau Manukan, Pulau Sulug.

Distribution in the world. Sabah, Sumatra (Clavareau, 1914; Kishimoto-Yamada, Takizawa & Mahadimenakbar, 2016).

Barcode Index Number (BIN). BOLD:ADH5352

Remarks. Found feeding on the family Zingiberaceae plants.

SUBFAMILY CASSIDINAE

Tribe CRYPTONYCHINI Weise 1911

Genus BrontispaSharp, 1904.

Refer to Appendix A for the generic diagnosis of this genus extracted from respective key literature of the taxa.

Brontispa longissima (Gestro, 1885)

(Fig. 11F)

Examined materials (39). Pulau Tiga: BOR/COL 8054–8062. Pulau Manukan: BOR/COL 8397, BOR/COL 9586–9589. Pulau Dinawan: BOR/COL 8453–8454, BOR/COL 8457, BOR/COL 8486–8491, BOR/COL 9751–9752. Pulau Mengalum: BOR/COL 9244–9248. Pulau Mamutik: BOR/COL 9610–9618.

Distribution in Sabah. Pulau Tiga, Pulau Manukan, Pulau Dinawan, Pulau Mengalum, Pulau Mamutik.

Distribution in the world. Bismarck Archipelago, Cambodia, Guangdong, Hainan, Hong Kong, Macau, Indonesia, Iran Jaya, Java, Moluccas, Nusa Tenggara, Sulawesi, Japan, Ryukyu Archipelago, Malaysia, Maldives, Myanmar, Philippines, Singapore, Taiwan, Thailand, Vietnam, American Samoa, Australia, French Polynesia, Nauru, New Caledonia, Papua New Guinea, Samoa, Solomon Islands, Vanuatu, Wallis and Futuna Islands (Kishimoto-Yamada, Takizawa & Mahadimenakbar, 2016; CABI, 2018).

Host plant. Areca catechu, Cocos nucifera, Elaeis guineensis, Metroxylon sagu, Phoenix roebelenii, Washingtonia robusta (CABI, 2018).

Barcode Index Number (BIN). BOLD:AAL7691

Intraspecific distance (%). Mean: 0.10  Max: 0.15

Remarks. Only found on the islands with tourisms activities and/or resorts, feeding on Cocos nucifera young shoot.

Tribe GONOPHORINI Weise 1911

Genus Gonophora Baly, 1858

Refer to Appendix A for the generic diagnosis of this genus extracted from respective key literature of the taxa.

Gonophora sp.

(Fig. 11G)

Examined materials (21). Pulau Tiga: BOR/COL 8016, BOR/COL 8519, BOR/COL 8532–8533, BOR/COL 9169, BOR/COL 9207–9209, BOR/COL 9770, BOR/COL 9803–9804. Pulau Gaya: BOR/COL 8260, BOR/COL 8266–8267, BOR/COL 9957, BOR/COL 9960. Pulau Sapi: BOR/COL 8344–8346. Pulau Sapangar: BOR/COL 8423–8424.

Distribution in Sabah. Pulau Tiga, Pulau Gaya, Pulau Sapi, Pulau Sapangar.

Barcode Index Number (BIN). BOLD:ADH6672

Intraspecific distance (%). Mean: 0.69  Max: 1.38

Remarks. Usually found on the leaf surface of Oncosperma tigillarium.

Tribe HISPINI Weise 1911

Genus Dactylispa Weise, 1897

Refer to Appendix A for the generic diagnosis of this genus extracted from respective key literature of the taxa. Two OTUs were identified and both OTUs were DNA barcoded.

Interspecific distance (%). Mean: 21.32  Max: 21.32

Dactylispa sp. 1

(Fig. 11H)

Examined materials (1). Pulau Tiga: BOR/COL 9777.

Distribution in Sabah. Pulau Tiga.

Barcode Index Number (BIN). BOLD:ADH5880

Remarks. Different from Dactylispa sp. 2 by the spine branching on the prothorax and smaller in size.

Dactylispa sp. 2

(Fig. 12A)

Examined materials (1). Pulau Gaya: BOR/COL 8305.

Distribution in Sabah. Pulau Gaya.

Barcode Index Number (BIN). BOLD:ADH6349

Remarks. Generally bigger in size than Dactylispa sp. 1.

Figure 12 Dorsal and ventral habitus of leaf beetle species.

(A) Dactylispa sp. 2; (B) Notosacantha sp. 1; (C) Notosacantha sp. 2; (D) Hispinae sp.; (E) Plagiodera sp.; (F) Phola sedecimpustulata; (G) Lema sp.

Tribe NOTOSACANTHINI Hincks 1952

Genus Notosacantha Chevolat, 1836

Refer to Appendix A for the generic diagnosis of this genus extracted from respective key literature of the taxa. Two OTUs were identified and both OTUs were DNA barcoded.

Interspecific distance (%). Mean: 14.14  Max: 14.14

Notosacantha sp. 1

(Fig. 12B)

Examined materials (4). Pulau Gaya: BOR/COL 8312, BOR/COL 9446–9448.

Distribution in Sabah. Pulau Gaya.

Barcode Index Number (BIN). BOLD:ADH5640

Remarks. Found on Ardisia sp. plant.

Notosacantha sp. 2

(Fig. 12C)

Examined materials (7). Pulau Tiga: BOR/COL 8540, BOR/COL 9771–9773, BOR/COL 9776. Pulau Mengalum: BOR/COL 9550–9551.

Distribution in Sabah. Pulau Tiga, Pulau Mengalum.

Barcode Index Number (BIN). BOLD:ADH5641

Remarks. Found on Ardisia sp. plant.

Hispinae sp.

(Fig. 12D)

Examined materials (1). Pulau Gaya: BOR/COL 9417.

Distribution in Sabah. Pulau Gaya.

Barcode Index Number (BIN). N/A

Remarks. Elytra dilated at side, regularly with four interstices of two regular rows of punctures.

SUBFAMILY CHRYSOMELINAE

Tribe CHRYSOMELINI Reitter 1912

Subtribe CHRYSOMELINA Chen 1936

Genus Plagiodera Chevrolat, 1837

Refer to Appendix A for the generic diagnosis of this genus extracted from respective key literature of the taxa.

Plagiodera sp.

(Fig. 12E)

Examined materials (1). Pulau Tiga: BOR/COL 8514.

Distribution in Sabah. Pulau Tiga.

Barcode Index Number (BIN). BOLD:ADH0536

Remarks. Found on plants near southeast mud volcano of Pulau Tiga.

Subtribe PHYLLOCHARINA Weise 1915

Genus Phola Weise, 1890

Refer to Appendix A for the generic diagnosis of this genus extracted from respective key literature of the taxa.

Phola sedecimpustulata (Stål, 1857)

(Fig. 12F)

Examined materials (1). Pulau Peduk: BOR/COL 9882.

Distribution in Sabah. Pulau Peduk.

Distribution in the world. Sabah, Peninsular Malaysia, Laos, Thailand, Vietnam (Mohamedsaid, 2004).

Barcode Index Number (BIN). BOLD:ADH6695

Remarks. Pronotum with three spots forming a triangular shape, elytra with nine yellow spots and one of the spots at the tip of the elytra.

SUBFAMILY CRIOCERINAE

Tribe LEMIINI Heinze 1962

Genus Lema Fabricius, 1798

Refer to Appendix A for the generic diagnosis of this genus extracted from respective key literature of the taxa.

Lema sp.

(Fig. 12G)

Examined materials (1). Pulau Gaya: BOR/COL 9393.

Distribution in Sabah. Pulau Gaya.

Barcode Index Number (BIN). BOLD:ADH6230

Remarks. Found after a shower rain near the Padang Point Restaurant at Pulau Gaya.

Conclusions

The notable number of leaf beetle species collected from a small portion of the island habitats in Sabah (∼3%) during the course of this study, indicates that many species have yet to be discovered and that these habitats should not be neglected in the process of investigating Bornean leaf beetle diversity. Moreover, the number of DNA barcodes generated in this study not only proves the effectiveness of barcoding in species delimitation, but also highlights the unsatisfactory level of representation of this taxa in the public sequence database for this region. This study marks an attempt to improve our current understanding of leaf beetle diversity from this region and helps in building up the DNA barcode reference database of these taxa.

Supplemental Information

Supplemental Information 1 BLAST and BOLD top-hit search results.

Click here for additional data file.

Supplemental Information 2 Barcode Index Number (BINs) Report.

Click here for additional data file.

Supplemental Information 3 Intraspecific distance.

Click here for additional data file.

Supplemental Information 4 Interspecific distance.

Click here for additional data file.

Supplemental Information 5 Appendix A. Generic diagnosis of genera.

Click here for additional data file.

Supplemental Information 6 BLAST top-hit search results.

Click here for additional data file.

Supplemental Information 7 Raw data for checklist.

Click here for additional data file.

We are grateful to the staff of various agencies for providing logistic support throughout the fieldwork: Justinus Guntabid, Sukur B. Sukardi, Muhammad Aliff B. Suhaimin, Simon Limbawang, Victor Siam and others (Sabah Park); Prof. Dr. Charles and ITBC staff, Assoc. Prof. Dr. Rossita Hj. Shapawi and IPMB boathouse staff (Universiti Malaysia Sabah); Marudu Express Travel Service staff (Pulau Dinawan); Mr. Balan and family (Pulau Sapangar). We appreciate assistance from Simon Kuyun, Foo She Fui, Phung Chee Chean, Phung Kin Wah, Jasrul Dulipat, and Choo Ming Huei during fieldwork. Special thanks to Foon Junn Kitt and Phung Chee Chean for manuscript checking.

Additional Information and Declarations

Competing Interests

Author Contributions

Field Study Permissions

DNA Deposition

Data Availability

The authors declare that they have no competing interests.

Kam-Cheng Yeong conceived and designed the experiments, performed the experiments, analysed the data, contributed reagents/materials/analysis tools, prepared figures and/or tables, authored or reviewed drafts of the paper, approved the final draft.

Haruo Takizawa analysed the data, contributed reagents/materials/analysis tools, authored or reviewed drafts of the paper, approved the final draft, species identification.

Thor-Seng Liew conceived and designed the experiments, analysed the data, contributed reagents/materials/analysis tools, prepared figures and/or tables, authored or reviewed drafts of the paper, approved the final draft, provides idea for data presentation.

The following information was supplied relating to field study approvals (i.e. approving body and any reference numbers):

Field experiments were approved by the Sabah Park for Tunku Abdul Rahman Park and Pulau Tiga Park, and

Sapangar Naval Base for Pulau Udar Kecil.

The following information was supplied regarding the deposition of DNA sequences:

DNA barcodes generated are accessible via DOI 10.5883/DS-BCHRY18.

The following information was supplied regarding data availability:

The raw data in available in the Supplemental Files and at Figshare (R script, raw data): https://figshare.com/projects/Investigating_leaf_beetles_Coleoptera_Chrysomelidae_on_the_west_coast_islands_of_Sabah_via_checklist-taking_and_DNA_barcoding/24061.

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
