# Peer review of "Investigating leaf beetles (Coleoptera, Chrysomelidae) on the west coast islands of Sabah via checklist-taking and DNA barcoding"

_PeerJ, doi:10.7717/peerj.5811_

## Round 0.1 · original submission · Major Revisions

Dear Dr. Yeong and colleagues:

Thanks for submitting your manuscript to PeerJ. I have now received three independent reviews of your work, and as you will see, the reviewers raised many concerns about the research. It is likely that your work would benefit from collaborating with an authority on DNA barcoding, with such an approach helping refocus the study on reporting a targeted compilation of DNA barcoding data for these beetles. A substantial concern of the reviewers is how this work fits into the larger picture. At present, your manuscript does not deliver on this, and so the overall relevance and impact on the field is lost.

Your manuscript also needs to be evaluated and edited by an English-speaking expert. The clarity in many places is lacking.

Please also address the many comments about experimental design, data collection, statistical analyses, explanation of protocols and approaches and improvements to overall presentation. Finally, the reviewers point out literature that seems to be missing from your cited references, as well as references that are not relevant or cited improperly.

I encourage you to substantially revise your work and resubmit to PeerJ. I look forward to your revision, and best of luck.

-joe

Reviewer 1 ·

Basic reporting

There are too many linguistic errors so the paper should be checked by an English native speaker.

Experimental design

The experimental design looks sound but lacks details. For instance, were the samples taken from disturbed habitats (secondary forests, plantations)? primary rain forest ? if yes, please add that information to the checklist and indicate whether there is any difference in species richness between different habitats?

Validity of the findings

The DNA barcoding analysis is well conducted and the presentation of the results is adequate but please see comments below:

1. The values of Table 2 on sequence nucleotide composition are shown again in the main text (Lines 184-190). I fin dit completely uninformative and unless these results are compared and discussed with other groups I would remove it.

2. I find Table 3 uninteresting. What does it show ? that using genbank is not the best way of identifying tropical leaf beetles ? Instead of hits to genbank I would show the BIN and ID of the nearest Neighbor and also the distance to Nearest Neighbor.

3. It would be interesting to have a table with the number of species/BINs per island and number of those that are shared with other islands and with the mainland.

4. The checlist only shows distribution in Sabah, but what is the geographical distribution worldwide ?

5. I wonder about the biogeographical composition of the fauna discovered. is exclusively oriental ? what are the 4 non-unique BINs already present in BOLD come from ? do they occur in a diffrent biogeographical region ? are there any cosmpolitan/invasive/alien species (Brontispa ?)?

6. These are important phytophagous isnects yet nothing is said about their host plants. Please add any information known on host plant use from literature and new host plant records gathered during fieldwork if any.

7. Photos of adult vouchers are already available in BOLD via the DOi number of teh data set, so I wonder whether it is worth including them in the manuscript

Additional comments

The study is very interesting because it focuses on an important hot spot of biodiversity and deals with a hyperdiverse group of insects. In addition, it contributes with a significant amount of novel DNA barcodes.

Reviewer 2 ·

Basic reporting

This study investigates leaf beetle diversity in 13 islands in Sabah area of Borneo. The species diversity was assessed based on morphological sorting and DNA barcoding.
The language of manuscript's main text appears fluent and good (though I am not a native English speaker), but the checklist part suffers from many small glitches (indicated below).

Introduction is quite specific and mostly deals with the history of leaf beetle research in Borneo. It would benefit much from a broader perspective to the topic. Why was this study conducted? Is it perhaps a part of a larger mission to investigate (leaf) beetle diversity in Borneo? Or globally? Please put your study to a broader context. There are large-scaler studies of tropical diversity (also in Borneo) that are not referred to. Is this study somehow related to any larger biodiversity project?

The figures are nice and informative. Table 2 does not contain, to my opinion, very relevant information and could be omitted (the meaning of these results are also not discussed).

The reference of Puillandre et al 2012 in line 234 appears incorrect. I cannot see how this reference is related to the text before it.

In the "checklist part", food plants are given for some, but not to all species. I wonder if this information was given for all taxa for which it was known.

Minor comments:
Line 203: Genbank > GenBank
Line 208: The dominant leaf beetle... > The dominant leaf beetles...
Line 275: both species not > both species have not been
Line 276: in GenBank probably > in GenBank are probably (verb missing)
Line 298: Differentiate from > Differentiated from
Line 554: Found resting on the beach Ipomoea > Found resting on the beach on Ipomoea
Line 634: Possible exhibit sexual > Possibly exhibits sexual
Similar small error are listed here are many more inthe checklist part.

Experimental design

The data is novel. Research questions are not very clearly stated and put to a larger context. Actually, in the introduction, no research questions are presented, but the authors just state that this study documents the species richness in 13 islands. It seems that large part of the observed species are new, non-described species, but this does not become very clear as the observed operational taxonomic units (OTUs) were identified by BLAST search only (?). This should be discussed more.

Material and methods should be more detailed. For example, it remains unclear how many plots were sampled in each island. Was handpicking of specimens conducted in the same plots? Were different plants in the plot systematically examined? How were the specimens "identified to the genus and species level" as most of the specimens remained un-named Do the author actually mean that the specimens were sorted into putative species (operational taxonomic units) this way? Was it really so that this sorting yielded in exactly the same delineation of OTUs as DNA barcodes, or were the final OTUs formed as a consensus of morphological sorting and DNA barcodes? It seems that the final delineation of specimens is exactly the one that was suggested by DNA barcodes, and it remain unclear of the morphological sorting conducted beforehand agreed completely with that.

The OTUs were identified based on the BLAST search (only?). I wonder why did not the authors use BOLD search, because BOLD has lots of records that are not present in GenBank and the data is cleaner (the authors acknowledge the problem of many misids in GenBank). BOLD identification tool would be a good option, and, importantly, all records (incl. unpublished) are available for identification. In the results section, the authors state: "including 68 species in 33 genera [was discovered]". However, I find that the authors should not speak about species at all, as they could not name the taxa. They should instead call them as putative species or OTUs. The authors also compiled a "species checklist", but it is not a true checklist as many OTUs were identified to the subfamily or genus level only. The inconsistent use of the term "species" is also reflected elsewhere. For example, on lines 226-228 the authors state: "...these five species are morphologically hard to distinguish from one another and thus, all these five species are...". Again, the authors should speak about putative species or OTUs. The use of the term "species" for the clusters should be revised throughout in the manuscript.

Validity of the findings

It seems that many new species were discovered, although this remains unclear, as it is not stated if the authors tried to get names for their OTUs in any other way but by BLAST search. The authors did little to assess/extrapolate how large part of the total leaf beetle diversity was discovered. The BLAST hits of less than 95% match often have little meaning. In some cases, the closest hit is a moth (Agrius convolvuli), in many other cases the closest hit is only to a subfamily level. Actually, only in a very few cases the BLAST search yielded really useful information. This calls for more discussion on the identities of the species, for example if the poor success is because the fauna of the region is so poorly represented in GenBank, or if this is because high levels of endemism in the region.

The conclusions are virtually lacking. Like introduction, the discussion part is quite specific. A broader scope here would be very beneficial for the whole paper.

Additional comments

This paper is with publication potential, but suffers mostly from a very narrow perspective to the topic. The paper would greatly benefit from wider perspective on tropical diversity and questions related to it. Also, the rationales for the study should be clarified. What motivated the authors to make this study? There are some weaknesses in the methodological part as well. They should be addressed. After a major revision, I believe that the paper would become acceptable for publication in PeerJ.

Reviewer 3 ·

Basic reporting

This is a complicated case. I spent one hour just digesting the abstract and introduction, and find that this manuscript would benefit from a thorough and careful editing / rewriting process. This I cannot deliver as a reviewer. For some issues, please refer to the "General comments for the author" section. I stopped reviewing the text after the end of the introduction for these reasons.

In general, I like this study as any advance in molecular biodiversity assessment is most welcome and urgently needed to build the global database needed to address the challenges of environmental degradation and ill directed political decisions.

However the current study lacks sound hypotheses and remains of very little relevance for a broader audience - with only 101 sequences from perhaps 68 species. Yes, the study is set on islands, but all of these were very small and very close to mainland, so that I also doubt that this work has particular relevance for island biogeography. It would certainly be interesting to see how many taxa were actually island endemics, I would suspect none. This would require in depth sampling of mainland areas.

Experimental design

See above, I think this is original primary research, yes, but needs to be much better presented and integrated in a larger scale framework. Without substantial sampling from the mainland, this remains a study with no obvious hypothesis testing or scientific question.

Validity of the findings

See above.

Additional comments

Language:
Overall, the paper needs substantial proofreading. I only provide some examples in my review, but this should help to understand where more work is needed.



Abstract:

The summary starts chaotic and needs to be rewritten.

Sabah is a province of Malaysia, and part of the island of Borneo. The current study looks into the leaf beetle fauna of a few very small satellite islands off the coast of Sabah.

"Sabah, northern Borneo is one of the world’s most well-recognized biodiversity hotspots" is a bit confusing and exaggerated a lot.

First of all, please study the definiton of biodiversity hotspot: https://en.wikipedia.org/wiki/Biodiversity_hotspot

I would write something like: Borneo is one of the World's truly megadiverse areas, yet its arthropod fauna remains largely unknown. Here, we present a study.....

"Plenty of studies...." better say: Several studies... Plenty is a lot and not appropriate wording here.

"In this study, we collected leaf beetle fauna from 13 islands..." better: We sampled leaf beetles from 13 islands....

"when the species names could not be determined" better: when the samples could not be identified to species level using the existing literature

"In addition, DNA barcodes – mitochondarial COI gene – of the species were sequenced." better: In addition, we generated DNA barcodes for the speciemens.

"A total of 68 species from 31 genera and 5 subfamilies were collected with 12 species name being determined." better:

"We collected 68 putative species from 31 genera and 5 subfamilies according to morphology, 12 of these could be identified to species level."

"In addition, DNA barcoding also reveals phenotypic variation
in leaf beetle species, particularly in the case of the subfamily Galerucinae"

Was that not known long before?

"This study provides baseline knowledge and information about the DNA barcodes of leaf beetle species on Sabah’s island habitats for use in future studies." This is not very appropriately in wording.

The study provides the first DNA barcoding data for leaf beetles from this part of the World, with the next steps being larger scale sampling across a wider geographic area in order to better understand large scale patterns of tropical arthropod diversity.



Introduction:

Generally, there are too many references.

"Although Borneo is recognized as one of the world’s biodiversity hotspots," see my comments above.

"taxonomic research on Borneo leaf beetles has been limited to a few publications," In the abstract it is said there were "plenty" of studies.

"possibly due to logistical difficulties and the inaccessibility of forest habitats" I am not entirely sure what this means. Yes, Bornean forests are in decline, but it should be really easy to local researchers to access forest, even in these days and time and even outside national parks I would suspect.

"Although Sabah has the highest number of islands in Malaysia, leaf beetle species diversity on islands has been little explored. In view of the fact that island habitats are generally known to have high species endemism"

Yes, but most of the islands studied here are very small and very close to the mainland. The most remote island visited here is Mengalum Island and only 50km offshore Kota Kinabalu. The scale in Fig 1A does not seem to be correct.

"To date, there are 73 records of leaf beetles with 15 Barcode Index Numbers (BINs) registered in the Barcoding of Life Data system (BOLD), but none of these records are from Sabah or Borneo."

I checked. http://www.boldsystems.org/index.php/Taxbrowser_Taxonpage?searchMenu=taxonomy&query=Chrysomelidae&taxon=Chrysomelidae

Specimen Records: 40,613
Public BINs: 2,034

"For all the reasons stated above, this study (1) documented the species richness of leaf
beetles from 13 selected islands on the west coast of Sabah, and (2) sequenced DNA barcodes of the leaf beetles to provide phenotypic polymorphism information and baseline DNA barcoding knowledge for future taxonomy study."

This conclusion of the introduction says all and nothing. I would suggest to also reword this to better reflect the aims of the study and hypotheses to be tested.

---

## Round 0.2 · accepted · Accept

Dear Dr. Yeong and colleagues:

Thanks for re-submitting your manuscript to PeerJ, and for addressing the concerns raised by the reviewers. I now believe that your manuscript is suitable for publication. Congratulations! I look forward to seeing this work in print, and I anticipate it being an important resource for chrysomelid workers. Thanks again for choosing PeerJ to publish such important work.

Best,

-joe

#